# The Role of Transabdominal Ultrasound in the Diagnosis of Early Stage Pancreatic Cancer: Review and Single-Center Experience

**DOI:** 10.3390/diagnostics9010002

**Published:** 2018-12-26

**Authors:** Reiko Ashida, Sachiko Tanaka, Hiromi Yamanaka, Suetsumi Okagaki, Keiko Nakao, Junko Fukuda, Miho Nakao, Tatsuya Ioka, Kazuhiro Katayama

**Affiliations:** Department of Cancer Survey and Gastrointestinal Oncology, Osaka International Cancer Institute, 3-1-69 Otemae, Chuo-ku, Osaka 541-8567, Japan; sachi686@cocoa.plala.or.jp (S.T.); yamanakahi@opho.jp (H.Y.); kanis@opho.jp (S.O.); nakaoke@opho.jp (K.N.); fukuda-ju@mc.pref.osaka.jp (J.F.); nakao-mi@mc.pref.osaka.jp (M.N.); ioka-ta@mc.pref.osaka.jp (T.I.); katayama-ka@mc.pref.osaka.jp (K.K.)

**Keywords:** US, pancreatic cancer, early cancer

## Abstract

Pancreatic cancer (PC) is the fourth leading cause of cancer-related death with a 5-year survival rate less than 10%. In the absence of effective screening methods, such as blood markers, most clinical diagnoses of PC are made at an advanced stage. However, early stage PC is associated with a more favorable five-year survival rate of 85.8% for stage 0, and 68.7% for stage IA. Transabdominal ultrasound (US) is frequently used as a first-line diagnostic tool in the clinical setting and a preferred modality for routine medical evaluations for asymptomatic individuals. Recently published Japanese data show that most PCs diagnosed in early stage had US findings, such as dilated main pancreatic ducts or pancreas cysts. For surveillance of high-risk individuals, such as those with an intraductal papillary mucinous neoplasm (IPMN), US is an ideal modality in terms of its non-invasive and cost-effective nature. However, the diagnostic performance of ultrasound varies greatly by the operator’s experience and the patient’s condition. This article reviews the present situation of early diagnosis of pancreatic cancer by US, along with tips for improving visualization of the pancreas.

## 1. Introduction

Pancreatic cancer (PC) is the fourth leading cause of cancer-related death worldwide, including Japan [1]. PC has an average annual incidence rate of 30.2 per 100,000 in men and 26.9 per 100,000 in women, which almost equals its mortality rate in 2014 in Japan [2]. The majority of pancreatic cancer presents in late stages at the time of diagnosis, which explains the poor associated prognosis. Although resectable disease has suboptimal survival rates, it is clearly more favorable with a five-year survival rate of 85.8% for stage 0 (Union for International Cancer Control, or IUCC, staging), 68.7% for stage IA and 59.7% for stage IB, according to the Japan Pancreatic Cancer Registry [3].

Whereas there have been great advances in the early detection and treatment of other malignancies, such as gastric cancer or colorectal cancer, developing screening systems for pancreatic cancer is still challenging due to a lack of early-stage biomarkers.

Transabdominal ultrasound (US) is frequently used as a first-line diagnostic tool for patients presenting with jaundice or abdominal pain, as it is a non-invasive and cost-effective modality. US is also a preferred modality for annual health or screening exams for asymptomatic individuals because it can be performed by a technician in Japan. However, the diagnostic performance of ultrasound greatly depends on the operator’s experience and the patient’s condition, such as obesity, and the presence and amount of interfering bowel gas. 

This paper will review the role of US for detecting early-stage pancreatic cancer and strategies to optimize visualization of the pancreas by US.

## 2. Current Situation of Early Diagnosis of Pancreatic Cancer by US

Ultrasound imaging has made significant advances in recent years and plays an important role in the detection, characterization, and staging of pancreatic diseases. US is a noninvasive imaging modality that remains the first diagnostic test when pancreatic cancer is suspected. However, the sensitivity and specificity of transabdominal ultrasound for pancreatic cancer range from 75% to 89%, and 90% to 99%, respectively [4]. The sub-optimal performance stems from several factors, including operator experience, large body habitus, the retroperitoneal location of the pancreas, and the presence of bowel gas.

Despite these limitations, US may play an important role in early detection of PC, as recently described by Kanno and colleagues [5]. In their multi-center retrospective study of 200 cases of early-stage pancreatic cancer (51 stage 0 disease and 149 stage 1 disease), US exams were performed in 135/200 (67.5%) cases, which showed main pancreatic duct (MPD) dilatation in 101/135 (74.8%) cases, MPD stenosis in 27/135 (20.0%) cases, and tumor detection in 71/135 (52.6%) cases. The presence of symptoms in 50 cases (25.0%), abnormalities on medical check-up in 34 cases (17.0%) and abnormalities during examination or follow-up for other diseases in 103 cases (51.5%) led to further medical examination. 

Of the 34 patients in whom abnormalities were detected during medical check-ups, 31 cases (91.2%) were detected by US. The abnormal findings detected were dilatation of the MPD in 21/31 (67.7%) cases, direct detection of a tumor in 9/31 (29.0%) cases, and stenosis of the MPD in 1/31 (3.2%) cases. Moreover, among the 103 patients in whom an abnormality was incidentally detected during examination or follow-up for other diseases, 41 (41.4%) cases had abnormal imaging findings on US. Among stage I PC cases, tumors were detected by US in 68/101 (67.3%) cases, by computerized tomography (CT) in 96/146 (65.8%) cases, by magnetic resonance imaging (MRI) in 73/127 (57.5%) cases and by endoscopic ultrasound (EUS) in 122/132 (92.4%). These results support an important role for US in detecting early-stage PC in Japan. 

However, according to a nationwide calculation of cancer screening detection rates, as reported by the Japanese Society of Ningen Dock, the number of pancreatic cancer cases discovered during Ningen Dock examinations (self-sponsored comprehensive examinations) in 2015 was only 100 cases among 1,794,817 male examinees (0.006%) and 52 cases of 1,210,576 female examinees (0.004%), which is not acceptable given the average annual incidence rate of PC in Japan [6].

In general, a technician performs the US examination and a doctor makes the judgment on the findings in a Ningen Dock examination. The standard examination time is about 6–7 min, in which it is necessary to describe the overall findings of the abdomen. Therefore, the time focused on the pancreas is at most 1–2 min, and it is difficult to observe the whole pancreas. Therefore, it is important to learn how to quickly identify indirect findings related to PC, such as pancreatic cysts and main pancreatic duct dilation. At the same time, it is more important to conduct the US examination using various techniques described below to optimize the sensitivity of US screening.

## 3. Pancreatic Anatomy and Tips to Improve Visualization by US

### 3.1. Anatomical Considerations of the Pancreas

The pancreas arises from the fusion of a ventral and a dorsal pancreatic bud during embryonic development. The dorsal pancreatic bud forms the head, neck, body, and tail, whereas the ventral pancreatic bud forms the uncinate process. The pancreas is usually located at the level of the first or second lumbar vertebra, although the location can vary depending on the phase of respiration. The pancreas head can be located in the sagittal axis, and the uncinate process originates from the pancreatic head and lies posteriorly to the superior mesenteric vessels. The pancreatic body and tail can be located in the transverse axis adjacent to the splenic hilum, although the shape varies greatly among individuals. The splenic vein runs from the splenic hilum along the posterior superior surface of the pancreas. The splenic artery arises from the celiac artery and runs slightly anterior along the superior margin of the pancreas.

### 3.2. Examination of the Pancreas by US

US should be performed routinely along multiple scan planes, including transverse, longitudinal, and angled oblique, to visualize the entire organ. It is also important to conduct the examination using various techniques, such as moving the transducer and applying compression to displace bowel gas, filling the stomach with liquid, examining the patient in suspended inspiration or expiration, and changing the patient’s position for better visualization of the pancreas.

For better visualization, our institute developed the special pancreatic US (pUS) protocol as shown below, which is dedicated to the visualization of the pancreato-biliary area since 1998 [7]. This protocol improves the detection rate of pancreas cystic lesions from 70.2% to 92.2% [8]. The sensitivity of detecting pancreatic cysts by special pancreatic US was 88.7% for the uncinate process and inferior head, 97.5% for the head, 97.1% for the body, 89.0% for the body-tail, and 66.7% for the tail. Whereas that of routine upper abdominal US was 74.2% for the uncinate process, 69.5% for the head, 81.0% for the body, 67.0% for the body–tail, and 26.7% for the tail (*p* < 0.001–0.016). 

### 3.3. Special Pancreatic US

#### 3.3.1. Premedication

It is ideal to perform US examination in the morning. The visualization will depend on the patient’s bowel condition and the food consumed the day before the examination. We recommend the patient to finish dinner before 9:00 p.m. and to consume less fibrous and easily digestible food. 

#### 3.3.2. Fowler Position and Body Position Change

For better visualization of the pancreas, Fowler’s position is ideal as the liver will descend toward pancreas side becoming an acoustic window. Moreover, gastric gas can be moved away from the scanning area due to gravity. One way to simulate Fowler’s position is to have the patient place their hands behind their back. However, this method can stiffen the abdominal wall, which can interfere with visualizing the pancreas. Moreover, it is very tough, especially for old patients to maintain this position for a long time. Therefore, the use of a special seat with a backrest that can electrically reposition a patient from the flat position to Fowler’s position is recommended (Figure 1).

Although the pancreas is thought to be a retroperitoneal organ, the pancreas is very mobile to changes in position. Therefore, it is very important to consider changing the patient’s position to better visualize the pancreas. For example, the right lateral decubitus position will help visualize a pancreatic body or tail lesion and the left lateral decubitus position for the pancreatic head, uncinate process, and biliary tract.

#### 3.3.3. Manual for Examination of the Pancreas

This is the manual for the examination of the pUS. At our institution, at least twelve pictures are taken according to the manual as shown Figure 2, which may minimize operator variability.

**A. Median Longitudinal Scan:** Visualize the celiac artery (CA) and superior mesenteric artery (SMA) longitudinally and check for lymph node swelling or main pancreatic duct dilation.

**B. Transverse Scan:** Rotate the probe into the transverse direction and visualize the aorta (Ao), celiac artery (CA), splenic artery (SpA) and common hepatic artery (CHA).

**C. Transverse Scan:** Check the splenic vein (SpV) and dorsal pancreas by visualizing the pancreatic body transversely as much as possible.

**D. Transverse Scan:** Enlarge the image and check the presence of main pancreatic duct (MPD) dilation by measuring its diameter at the pancreatic body. It appears as a thin hypoechoic line bordered by two echogenic margins. The upper limit of the main duct is 2.5 mm at the body.

**E. Transverse Scan:** Observe the pancreatic head by shifting the probe toward the right side of the patient. Follow the MPD from the pancreatic body to the ampulla as much as possible and measure the duct size of the MPD at the head.

**F. Right Subcostal Longitudinal Scan:** Rotate the probe toward the longitudinal position and observe the lower part of the pancreatic head and uncinate as well as the portal vein and inferior vena cava (IVC).

**G. Right Subcostal Margin Scan:** Detect the gallbladder longitudinally and check the size of the gallbladder and the existence of abnormalities, such as stones or debris by moving the patient into the left lateral decubitus position.

**H. Right Subcostal Longitudinal Scan:** Observe the hilar hepatic area, portal vein, common bile duct, and superior aspect of the pancreatic head. Check for bile duct dilation by moving the patient into the left lateral decubitus position.

**I. Right Subcostal Longitudinal Scan:** Observe the pancreatic head and check for MPD narrowing or common bile duct (CBD) stricture by following the bile duct, which is done by moving the patient into the left lateral decubitus position.

**J. Left Subcostal Longitudinal-Oblique Scan:** Visualize the pancreatic body to tail in the longitudinal plane by shifting the probe from the pancreatic body to tail, along the splenic vein.

**K. Left Intercostal Scan:** View the pancreatic tail by detecting the splenic hilar, located at the dorsal side of the splenic vein, by moving the patient into the right lateral decubitus position.

**L. Left Subcostal Tilting Scan and The Liquid-Filled Stomach Method:** Screen the pancreatic body to tail thoroughly using the liquid-filled stomach method. The pancreatic body to tail can be visualized more clearly by moving the patient into the right lateral decubitus position.

#### 3.3.4. The Liquid-Filled Stomach Method

The liquid-filled stomach method can improve visualization of the pancreas [9,10]. The patients drink about 100 to 300 mL of degassed water, creating a gastric sonic window by simply displacing gas to the fundus of the stomach.

At our institution, the liquid-filled stomach method with Fowler’s position is usually added after conventional screening of the pancreas. This method, in particular, improves the visualization of the pancreatic body and tail by eliminating gastric gas. (Figure 3) This method also helps visualize the pancreatic uncinate and ampulla by enhancing the contrast between the target and intestine. 

At our institution, commercially-available black tea with milk is used for the liquid-filled stomach method [8]. For the individual with diabetes mellitus, or who cannot tolerate milk tea, clear green tea is used instead. The liquid needs to be served in steel cans or heat-resistant polyethylene terephthalate bottles to avoid air contamination. A straw is recommended to drink the liquid to avoid the air ingestion and patients are encouraged to belch to reduce any residual air in the stomach. The liquid-filled stomach method is not helpful for patients after gastric surgery, or evaluation in the flat position, because the liquid does not stay in the proper stomach location to improve visualization. 

#### 3.3.5. High-Frequency Probe

Discrimination of the lesion, such as a pancreatic cyst or solid lesion, is sometimes difficult, especially when small. Moreover, artifacts such as multiple reflection echoes or debris sometimes make it difficult to make a correct diagnosis. In such cases, using a high-frequency probe helps to gain a clearer image by locating the lesion close to the surface as much as possible, and by using positional changes or breathing methods. 

### 3.4. Contrast-Enhanced US and Elastography

The role of real-time imaging using contrast-enhanced ultrasonography (CE-US) in pancreatic tumors has been reported since the late 1990s [11]. Oshikawa et al. reported the usefulness of CE-US in the differential diagnosis of pancreatic tumors by comparing it to dynamic CT in 2001 [12]. In 2014, D’Onofrio et al. conducted a meta-analysis regarding the diagnostic utility of CE-US for the differentiation of ductal adenocarcinoma from other pancreatic diseases, which showed a pooled sensitivity and pooled specificity of 0.89 (95% confidence interval (CI), 0.85–0.92) and 0.84 (0.77–0.89), respectively [13]. 

Kitano et al., used coded phase inversion harmonic ultrasonography and demonstrated a higher sensitivity of this technique, compared with contrast-enhanced CT, but similar to EUS for detecting lesions ≤2 cm [14]. Fan et al. also showed the superiority of contrast-enhanced ultrasonography compared to multiple detector computerized tomography (MDCT) for small- or medium-size pancreatic lesions [15]. Zhu et al. recently reported that double contrast-enhanced ultrasound (DCEUS) improved the detection and localization of occult lesions in the pancreatic tail [16]. Both intravenous microbubbles and oral contrast agents were used for DCEUS, showing sensitivity and specificity for depicting occult lesions ≤2.2 cm of 92% and 95%, respectively.

Elastography is a new imaging modality, which evaluates the stiffness of organs. Currently, there are two types of elastographies based on different principles, one is strain elastography and the other is shear wave elastography. Park et al. showed that shear wave elastography can determine the relative stiffness between a lesion and the background pancreatic parenchyma, which is helpful in differentiating benign and malignant solid pancreatic lesions [17]. Zaro et al., also showed that shear wave elastography can indicate an increase of elasticity in the pancreatic tumor, compared to normal pancreas [18]. Although there are multiple challenges to be solved, such as operator dependency or inadequate reproducibility, elastography may contribute to early detection of pancreatic cancer in the future.

### 3.5. Real-Time Virtual Sonography

Real-time virtual sonography (RVS) is a new method to synchronize US images with other imaging modalities, such as CT or MRI. The volume data of CT or MRI are pre-stored into the US machine and the real-time ultrasound image is displayed simultaneously while the virtual view is reconstructed as a multiplanar reconstruction (MPR) image from the stored volume data. This technique helps operators have real-time recognition of the organ, even though the US image does not clearly visualize it due to bowel gas or other limitations. 

RVS technique will especially help the trainee to quickly understand the anatomy, as well as avoid the risk of missing a lesion. Although this method has been used for liver disease, especially during navigation in ultrasound-guided treatment, Sofuni et al., showed the utility of RVS in pancreato-biliary disease as well [19].

For pUS, it was difficult for first-generation RVS systems to visualize the image correctly when the patients assume Fowler’s position, due to the change in position. Next-generation RVS systems will adjust for differences between the flat and Fowler’s positions. Therefore, it may be useful even during pUS. 

## 4. Importance of US in Regional Networks

In order to diagnose early-stage pancreatic cancer, it is important to maintain a high degree of suspicion for pancreatic cancer at the first visit. If patients present with symptoms such as epigastric pain, back pain, body weight loss and worsening of diabetes, US needs to be performed proactively in the clinic. If indirect findings of pancreatic cancer, such as main pancreatic dilation or pancreatic cyst are visualized, it is important to refer them for further examination. 

Hanada et al. have advocated for the importance of developing regional networks between specialists in pancreatic disease and general practitioners (GP) [20]. They established a community program between Onomichi General Hospital and Onomichi Medical Association to find early-stage pancreatic cancer since 2007. Hanada et al. educated GPs about the risk factors of PC, such as IPMN, chronic pancreatitis, diabetes mellitus (DM), or a familial background of PC. They also encouraged GPs to perform US to look for even slight changes in the pancreas, such as main pancreatic dilation or pancreatic cysts, to then refer them to the Onomichi General Hospital for further examination. 

From January 2007 to June 2014, a total of 6475 cases were referred to the hospital since starting this program. As a result, 399 out of 6475 cases (6.2%) were histologically diagnosed as PC, where 16 cases were PC in situ. Currently the value of these programs is widely recognized in Japan and hopefully, the regional networks system will be developed across Japan. 

## 5. Surveillance of High-Risk Individuals Using Pancreatic US

In our department, we have clarified that main pancreatic duct dilation and cystic lesions by US are independent risk factors of pancreatic cancer [21]. We hypothesized that early stage pancreatic cancer can be found efficiently by focusing on high-risk individuals (HRI) and conducting periodic surveillance. 

Because a minimally invasive and highly accurate method is desirable for surveillance, pUS is thought to be the best modality, as there is no radiation exposure and pUS was superior to simple CT to pick up cysts and pancreatic ductal dilations in our previous study [22]. 

For this reason, our department has conducted periodic surveillance for individuals with pancreatic cysts or a dilated main pancreatic duct every 3–6 months using pUS since 1998. So far, 1058 people are included from 1999 to 2002. During a mean follow-up of 75.5 months, pancreatic cancer subsequently developed in 12 of 1058 subjects (1.13%) [23]. Among them, three cases were stage 0, two cases were stage IA, and one case was stage IB. Our study also revealed that individuals with both a pancreatic cyst and a dilated MPD had a 27 times higher incident rate of developing pancreatic cancer, with 5.62% over five years.

In general, US is thought to be unsuitable for surveillance of HRI due to its low visibility, especially in Western countries. Many international guidelines for pancreatic cystic neoplasm, such as the Fukuoka guidelines [24], European guidelines [25] or American Gastroenterological Association (AGA) guidelines [26], do not include US as a modality for surveillance. Therefore, the role of US in HRI surveillance is still limited as long as the conventional method is utilized. 

In our department, pUS is used as a surveillance modality only when pUS can visualize at least 3/5 parts of the pancreas clearly. Annual CT or MRI is also added to cover the limitations of visibility as a study protocol. However, it is not recommended to use pUS for the surveillance of individuals with large oligo-locular IPMN (>3 cm), especially in the pancreatic head because of the echo attenuation, individuals who are status post-gastric surgery because the liquid-filled stomach method does not improve the visibility, and those individuals with multiple cysts (i.e. more than 10) due to time. These HRIs are good candidates for EUS. In contrast, young HRIs who are thin and healthy but have a strong gag reflex during EUS are good candidates for pUS, as well as old patients who have are at relatively high-risk for endoscopic procedures. 

Although a pUS-based system may not be as sensitive compared to an EUS-based surveillance system [27], the latter is less available. A surveillance system using pUS therefore, is thought to be more useful and feasible for the early detection of pancreatic cancer. Hopefully, this approach with regional networks will be more accepted in the future.

## 6. Summary

The current state of abdominal ultrasound in the early diagnosis of pancreatic cancer was discussed. The morbidity rate of pancreatic cancer will increase as the society ages in Japan. As a noninvasive examination and with wide availability of technicians trained to use standardized screening approaches, US has significant potential to impact early detection efforts. We hope that many institutions will apply this pancreatic US protocol to examine the pancreas as a tool to diagnose early-stage pancreatic cancer.

## Figures and Tables

**Figure 1 diagnostics-09-00002-f001:**
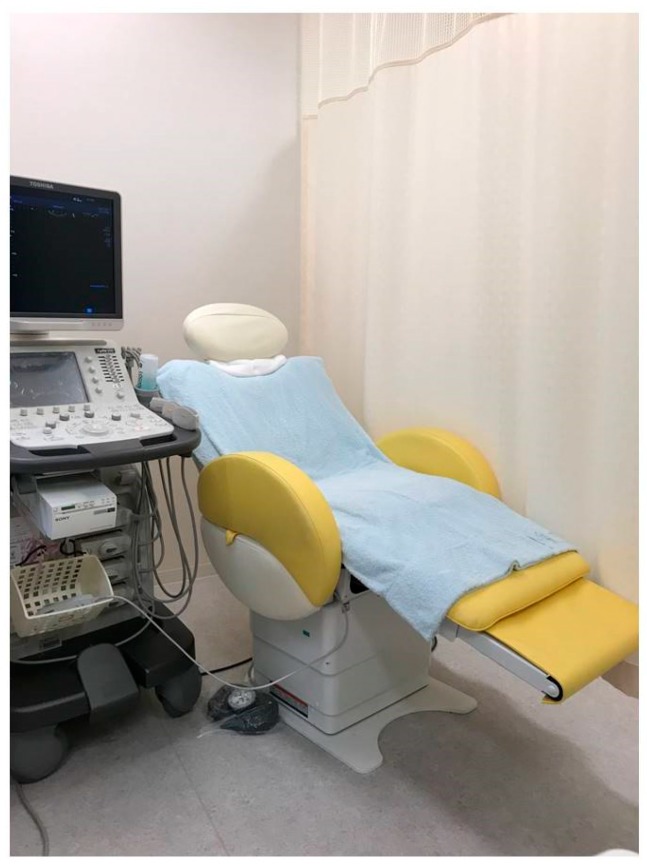
Examination table with an adjustable backrest during special pancreatic ultrasound (US).

**Figure 2 diagnostics-09-00002-f002:**
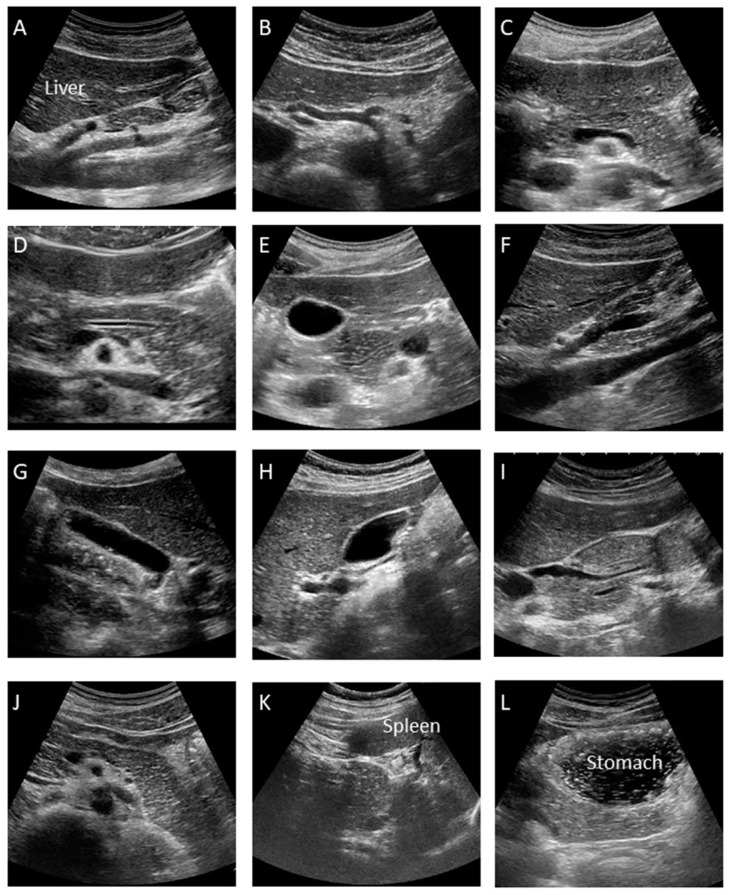
Standard recording sites during special pancreatic US (12 images). (**A**) Median Longitudinal Scan. (**B**) Transverse Scan. (**C**) Transverse Scan. (**D**) Transverse Scan. (**E**) Transverse Scan. (**F**) Right Subcostal Longitudinal Scan. (**G**) Right Subcostal Margin Scan. (**H**) Right Subcostal Longitudinal Scan. (**I**) Right Subcostal Longitudinal Scan. (**J**) Left Subcostal Longitudinal-Oblique Scan. (**K**) Left Intercostal Scan. (**L**) Left Subcostal Tilting Scan and The Liquid-Filled Stomach Method.

**Figure 3 diagnostics-09-00002-f003:**
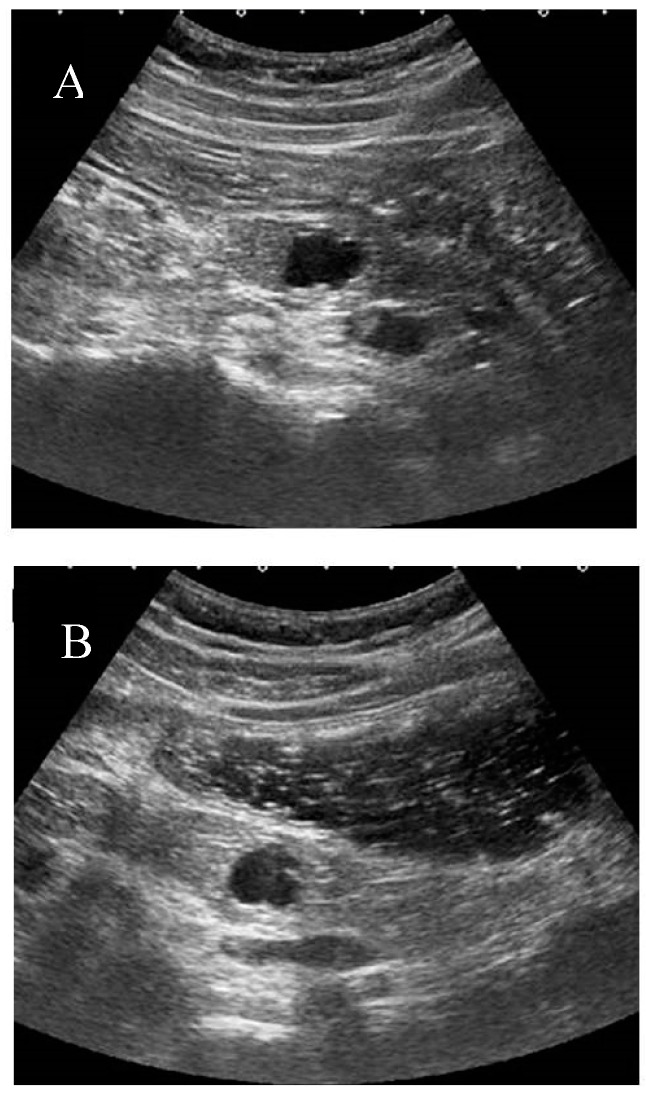
The liquid-filled stomach method. The pancreas tail is not well visualized due to gastric gas in normal examination (**A**). However, the pancreatic body to tail become clearly visualized after intake of milk tea with Fowler’s position (**B**).

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
