# Peer review of "The Role of Transabdominal Ultrasound in the Diagnosis of Early Stage Pancreatic Cancer: Review and Single-Center Experience"

_diagnostics, 2018, doi:10.3390/diagnostics9010002_

Reviewer 1 Report

The review is of excellent quality. I would suggest some minor changes:

1. Please describe in detail what kind of patients should be subjected to special pancreatic US in your institution. Please describe if there is a target group especially strongly recommending special pancreatic US.

2. Please attach a number to each echo image in Figure 2 to match the description number in the text.

3. In the contrast enhanced US section, recently, Zhu et al. has described that double contrast-enhanced ultrasound (DCEUS) improved the detection and localization of occult lesions in the pancreatic tail. According to the study, the sensitivity and specificity of DCEUS for depicting occult lesions of ≤ 2.2 cm were 92% and 95%, respectively. Please cite this article.

Zhu W, Mai G, Zhou X, Song B. Double contrast-enhanced ultrasound improves the detection and localization of occult lesions in the pancreatic tail: a initial experience report. Abdom Radiol (NY). 2018 doi: 10.1007/s00261-018-1746-2

4. The usefulness of pancreatic tumor diagnosis using elastography (including shear wave elastography) has also been reported, and in the future it may be a useful modality for the detection of early pancreatic cancer. Please also refer to the following articles on elastography in pancreatic tumor diagnosis.

Park MK, Jo J, Kwon H, et al. Usefulness of acoustic radiation force impulse elastography in the differential diagnosis of benign and malignant solid pancreatic lesions. Ultrasonography 2014;33:26-33.

Zaro R, Dina L, Pojoga C, et al. Evaluation of the pancreatic tumors by transabdominal Shear Wave Elastography: preliminary results of a pilot study. Med Ultrason. 2018;20:285-291

5. Please mention in the Discussion part if there is a suggestion on how to use transabdominal US and EUS.

Author Response

1. Because our institution is cancer center, all patients are referred due to some concern for cancer such as elevation of tumor marker, pancreatic lesion including tumor, cyst or dilated MPD. We use pUS for all these patients as a first screening modality at the initial visit. We recommend to use pUS for all patients suspected of having pancreatic disease as long as the patients are not obese.

However, as a surveillance modality, we restricted cases using pUS. We add the sentence below in the section of surveillance.

In our department, pUS is used as a surveillance modality only when pUS can visualize at least 3/5 part of the pancreas clearly. Annual CT or MRI is also added to cover the limitation of visibility as a study protocol. However, it is not recommended to use even pUS for the surveillance of individual with large oligo-locular IPMN (>75px) especially in the pancreatic head because of the echo attenuation, individuals who are status post gastric surgery because the liquid-filled stomach method does not improve the visibility, and those individuals with multiple cysts (ex. more than 10) due to time. These HRIs are good candidates for EUS.

2. We add the number in the Figure 2.

3. We add the paper in the section of contrast EUS. We changed the sentence like below.

Zhu et al. has recently reported that double contrast-enhanced ultrasound (DCEUS) improved the detection and localization of occult lesions in the pancreatic tail.16 Both intravenous microbubbles and oral contrast agent were used for DCEUS showing a sensitivity and specificity for depicting occult lesions ≤ 2.2 cm of 92% and 95%, respectively.

4. We add the paper in the section of contrast EUS and Elastography. We changed the sentence like below.

Elastography is a new imaging modality which evaluates the stiffness of organs. Currently, there are two types of elastographies based on different principles, one is strain elastography and the other is shear wave elastography. Park et al. showed that shear wave elastography can determine the relative stiffness between a lesion and the background pancreatic parenchyma, which is helpful in differentiating benign and malignant solid pancreatic lesions. 17 Zaro et al. also showed that shear wave elastography can indicate an increase of elasticity in the pancreatic tumor compared to normal pancreas.18  Although there are multiple challenges to be solved such as operator dependency or inadequate reproducibility, elastography may contribute to early detection of pancreatic cancer in the future.

5. The answer for this suggestion may overlap to the first suggestion. We add the sentence like below.

In our department, pUS is used as a surveillance modality only when pUS can visualize at least 3/5 part of the pancreas clearly. Annual CT or MRI  is also added to cover the limitation of visibility as a study protocol. However, it is not recommended to use pUS for the surveillance of individuals with large oligo-locular IPMN (>75px) especially in the pancreatic head because of the echo attenuation, individuals who are status post gastric surgery because the liquid-filled stomach method does not improve the visibility, and those individuals with multiple cysts (ex. more than 10) due to time. These HRIs are good candidates for EUS. In contrast, young HRIs who are thin and healthy but have a strong gag reflex during EUS are good candidates for pUS as well as old patients who have are at relatively high risk for endoscopic procedures. 

Reviewer 2 Report

Ashida R et al present a well written review about the role of abdominal ultrasound (US) in the diagnosis of early stage pancreatic cancer. Authors show in detail their own protocol regarding the use of the special pancreatic ultrasound, showing the technical details and some data

Interesting topic that can potentially improve early detection of pancreatic cancer. I congratulate the authors on their US technique and encourage on continue to disseminate the technique

Unfortunately the standard technique of  US worldwide does not use the presented protocol. This is a single center experience and as such the tittle should be changed to include that this is a single center protocol/experience. US is operator depended and as mentioned by authors, it can be influenced by the amount of gas and fat. Regarding the use of ultrasound in the surveillance of pancreatic cysts, there are many guidelines (Fukuoka, AGA, ACG, European, radiology) on the management and surveillance of pancreatic cysts and US is not part of the screening and surveillance of cysts. Please add some data regarding the most common and updated guidelines for the management of cysts 

I recommend the authors to soften the statements regarding the use of US as a screening tool, since it is still limited.

Author Response

Thank you for your kind comment and thoughtful suggestion. As you mentioned, US is not the standard modality for surveillance of HRI. We add relevant references and text such as noted below. We also changed the title.

The Role of Transabdominal Ultrasound in the Diagnosis of Early Stage Pancreatic Cancer ~Review and Single Center Experience~

In general, US is thought to be unsuitable for surveillance of HRI due to its low visibility especially in Western countries. Many international guidelines for pancreatic cystic neoplasm such as Fukuoka guideline24, European guideline25 or AGA guideline.26 do not include US as a modality for surveillance. Therefore, the role of US in surveillance of HRI is still limited as long as the conventional method is utilized.

24.  Tanaka M, Fernandez-Del Castillo C, Kamisawa T, et al. Revisions of international consensus Fukuoka guidelines for the management of IPMN of the pancreas. Pancreatology 2017;17:738-753.

25.  European evidence-based guidelines on pancreatic cystic neoplasms. Gut 2018;67:789-804.

26.  Vege SS, Ziring B, Jain R, et al. American gastroenterological association institute guideline on the diagnosis and management of asymptomatic neoplastic pancreatic cysts. Gastroenterology 2015;148:819-22; quize12-3.